# Terpene Profiles Composition and Micromorphological Analysis on Two Wild Populations of *Helichrysum* spp. from the Tuscan Archipelago (Central Italy)

**DOI:** 10.3390/plants11131719

**Published:** 2022-06-29

**Authors:** Lorenzo Marini, Enrico Palchetti, Lorenzo Brilli, Gelsomina Fico, Claudia Giuliani, Marco Michelozzi, Gabriele Cencetti, Bruno Foggi, Piero Bruschi

**Affiliations:** 1Department of Agriculture, Food, Environment and Forestry, DAGRI, University of Florence, Piazzale delle Cascine 18, 50144 Firenze, Italy; enrico.palchetti@unifi.it (E.P.); piero.bruschi@unifi.it (P.B.); 2Institute for the BioEconomy, National Research Council, CNR-IBE, Via Giovanni Caproni 8, 50145 Firenze, Italy; lorenzo.brilli@ibe.cnr.it; 3Department of Pharmaceutical Sciences, University of Milan, Via Mangiagalli 25, 20133 Milano, Italy; gelsomina.fico@unimi.it (G.F.); claudia.giuliani@unimi.it (C.G.); 4Institute of Bioscience and BioResources, National Research Council, CNR-IBBR, Via Madonna del Piano 10, 50019 Sesto Fiorentino, Italy; marco.michelozzi@cnr.it (M.M.); gabriele.cencetti@cnr.it (G.C.); 5Department of Biology, BIO, University of Florence, Via La Pira 4, 50121 Firenze, Italy; bruno.foggi@unifi.it

**Keywords:** *H. italicum*, *H. litoreum*, chemotaxonomy, SEM, glandular *indumentum*, bark terpene

## Abstract

Two wild populations of *Helichrysum* (Mill.) located at Elba Island (Tuscan Archipelago, Central Italy) were morphologically and phytochemically analyzed to taxonomically identify *H. litoreum* (population A) and *H. italicum* subsp. *italicum* (population B). Micromorphological and histochemical analyses were performed on the *indumentum* using Scanning Electron Microscope (SEM) and Light Microscope (LM). Morphometric analyses on vegetative and reproductive traits were also conducted. Finally, a chemotaxonomic analysis was carried out on the terpene profile of flowers, leaves and bark tissues using gas chromatography (GC-MS). Results suggested that morphological discriminant traits were mainly in leaves and cypselae glandular tissues. Phytochemical analysis indicated that a high relative content of α-pinene and β-caryophyllene were the main markers for population A, while a high relative content of neryl-acetate, α-curcumene, isoitalicene and italicene, especially in the terpene profile of bark tissue, were the main compounds for discriminating population B. The analysis suggested that the wild population A could be mainly ascribed to *H. litoreum*, whilst population B is defined by *H. italicum*.

## 1. Introduction

The genus *Helichrysum* Mill. belongs to the Asteraceae family and includes over six hundred species and subspecies globally widespread [1,2]. In the Mediterranean area, there are currently about 25 species [3] some of which have very similar genetic traits despite the fact that they may show different morphological characteristics.

These conditions make critical the correct taxonomic identification of *taxa* such as *Helichrysum italicum* subsp. *italicum* (Roth) G. Don and *Helichrysum litoreum* Guss. (≡ *H. angustifolium* (Lam.) DC.), which are very closely taxonomically related [4,5] *H. italicum* subsp. *italicum* and *H. litoreum* are perennial, xerophilous and suffruticose plants [4] growing over calcareous and siliceous substrates in arid, stony, shallow soils [6]. They can show very similar morphological traits, among which is a synflorescence composed of a few tens to up a hundred gold-yellow *capitula*, with similarities in shape and size [4]. However, a few discriminating characters can be observed. For instance, *H. italicum* subsp. *italicum* shows a glandular *indumentum* composed of secretory glands with a high density on leaf abaxial surfaces and flower bracts [4,7]; these structures are responsible for the essential oil production, which make it an aromatic plant with medicinal properties such as antimicrobial, anti-inflammatory, and anti-viral activities [8,9,10,11,12,13,14,15]. By contrast, *H. litoreum* exhibits a lower number of and sparser secretory glands compared to *H. italicum* subsp. *italicum*, thus resulting in a weakly aromatic or non-aromatic plant [4,6]. With regards to the ecological requirements, *H. italicum* subsp. *italicum* grows over a wider altitudinal range (0–2000 m a.s.l.) than *H. litoreum* (0–900 m a.s.l.), and it is recognized as a pioneer plant of rocky habitats [4,6], whilst *H. litoreum* is endemic to the Tyrrhenian region and prefers areas exposed to marine aerosols [6].

Currently, the most critical issues for the detection of the taxonomic boundaries between *H. italicum* subsp. *italicum* and *H. litoreum* came from the difficulty of distinguishing between the overlapping sets of morphological characters, which may result from environmental adaptation or introgressive processes. Despite genetic analysis remaining the most promising approach to solving the most critical taxonomical issues [1,16,17,18,19], chemotaxonomic analysis based on secondary metabolite profiles can help in the classification of plants [20] and several studies have analyzed terpene profiles of bark tissues to discriminate between different species and subspecies of crops and trees [21,22,23]. In particular, essential oil terpenes have proven to be very important chemotaxonomic markers to discriminate between species, hybrids, provenances and clones, since these compounds are under strong genetic control, and little influenced by environmental factors [24,25,26,27]. Many chemotaxonomic studies on *Helichrysum* spp. investigated the chemical composition of the essential oil from inflorescences [28,29,30,31,32] while others focused on the leaf volatilome [33,34]. Therefore, it is important to analyze the *Helichrysum* spp. terpene profile of the bark tissue to investigate the independence of the phytochemical composition from environmental factors and thus validate its use in chemotaxonomy. Furthermore, studies that addressed micromorphological analysis proved the taxonomic significance of several floral traits [4] and cypselae characteristics [35] for the discrimination between some species of *Helichrysum* spp.

In this framework, the primary objective of this work was to distinguish two wild populations of *Helichrysum* spp. located in Elba Island (Tuscan Archipelago, Central Italy) combining a morphological and a chemotaxonomic approach. These two populations are located at Capo d’Enfola (site A) and Monte Capanne (site B) and, based on morphological traits, were classified as belonging to *H. litoreum* and *H. italicum* subsp. *italicum,* respectively [36,37]. Specifically, a morphometric analysis of both the vegetative and reproductive plant parts and micromorphological and histochemical observations on the *indumentum* was carried out. Then, terpene analysis (monoterpenes and sesquiterpene profiles) of flowers, leaves and bark tissues was performed. Finally, all the investigated features were assessed as possible taxonomical markers to discriminate between the two wild populations of *Helichrysum* spp.

## 2. Results

### 2.1. Morphological Analysis

The results from ANOVA (Table 1) revealed that the most significant traits, which best differentiated between the two populations, were found in leaves (ID02, ID03, ID04, ID20, ID22), capitula (ID07, ID09, ID27), outermost bracts (ID10, ID11), innermost bracts (ID13, ID14), pappus (ID15) and cypselae (ID17, ID18, ID19, ID26).

The PCA analysis (Figure 1) indicated that the first two principal components explained about 51% of the total variation among the assessed characters (Appendix A). The two populations were separated along with the first principal component (Dim1). Variables describing leaf size (ID03, ID04), cypselae size and gland density of duplex hairs on cypselae (ID08, ID09. and ID17), and *capitula* width (ID09) were the most decisive traits and contributed to Dim1 (0.78, 0.40, 0.57, 0.77, 0.72, and 0.55, respectively), explaining 37% of the total variation. The second component (Dim2) accounted for 14.1% of the total variation and differentiated individuals by outermost bract size (ID10 and ID11; 0.65 and 0.55 on Dim2).

### 2.2. Micromorphological Analysis

In both populations, non-glandular and glandular trichomes present in the vegetative and reproductive organs were assessed (Appendix A; Figure 2 and Figure 3). The non-glandular trichomes were multicellular, simple, uniseriate, long filamentous hairs which were slightly twisted (Appendix A). The glandular *indumentum* was composed of multicellular and biseriate trichomes belonging to two main morphotypes (Appendix A). The club-shaped trichomes were composed of 10 (−12) cells arranged in two rows of five, or six, cells each: two basal cells, two stalk cells, from six to eight secretory cells of the club-shaped head surrounded by a subcuticular chamber storing secondary metabolites of terpenoid nature (Appendix A). The duplex trichomes were composed of two oblong cells, paired along their major axis, with the cuticular layer closely adhering to the hair surface and producing muco-polysaccharidic components (Appendix A); the distinction between the two cells of the duplex trichomes was more evident in the samples from population B (Monte Capanne) than in the samples from the population A (Capo d’Enfola).

The distribution pattern of trichome morphotypes in leaves over the two *Helichrysum* populations analyzed through SEM indicated that the leaves of all samples were covered by long filamentous non-glandular hairs on the abaxial and adaxial surfaces (Figure 2a–f). These hairs formed a dense *indumentum* that completely obscures the epidermises, making complex the sample preparation for light microscopy investigations. These appendages were responsible for the grey-silver appearance of the leaves, especially on the abaxial side.

The club-shaped glandular trichomes were present on both leaf laminas (Figure 2c,f), being more abundant on the abaxial surfaces in the interveinal regions, whereas they were scarce on the adaxial surfaces.

The distribution pattern of trichome morphotypes in *capitula* and cypselae analyzed in both *Helichrysum* populations using SEM indicated that club-shaped hairs occurred on the tips of the corolla lobes, both on the primordia and the fully developed florets in both populations (Figure 3a–e, arrows). The duplex glandular trichomes were present on the ovary surfaces (Figure 3b–f) and therefore on the cypselae surface; they showed a higher density in the population B samples in comparison to the population A ones (Figure 3c–g).

The SEM micrographs at higher resolution provided differences at a cell-surface level. The epidermis of cypselae of both populations showed cells organized in simple rectangular reticulate, with regular or curvilinear margins. However, clear differences between the two populations were observed on the outer surface of the cells. Specifically, the outer surface of the cells of cypselae collected at Capo d’Enfola (A) was densely covered with epicuticular waxes in the form of small papillae (Figure 3d). By contrast, the outer surface of the cells of cypselae collected at Monte Capanne (B) lacked epicuticular waxes (Figure 3h).

### 2.3. Terpene Analysis

Terpene data for bark, flowers, and leaves tissues of the two *Helichrysum* populations are reported in Appendix A A total of 36 compounds were identified and included 22 monoterpenes (Figure 4) and 14 sesquiterpenes (Figure 5).

#### 2.3.1. Monoterpenes

Among all 22 monoterpenes, 19 were found in flower tissue (82.6%), 20 in leaf tissue (87%), and 13 in bark tissue (56.5%).

In flower tissues, the most abundant and highly significant compounds were α-pinene, limonene, γ-terpinene and neryl-acetate. The α-pinene showed an average relative amount higher in population A (58.6%) than B (22.1%). Limonene, γ-terpinene and neryl-acetate were, on average, higher in population B (51.4%, 11.5%, 42.8% respectively) than A (18.3%, 1.7%, 1.9%). Furthermore, the presence of 2-bornene was detected only in the floral tissues in trace amounts in both populations (~0.25%).

In leaves, the most abundant and highly significant compounds were α-pinene, limonene, (E)-β-ocimene and neryl-acetate. The α-pinene showed a higher relative content in population A (55.1%) than B (14.6%). Limonene, (E)-β-ocimene and neryl-acetate were higher (47.8%, 11.3%, 33.9%, respectively) in population B than A (30.8%, 0.4%, 2.3%). (E)-β-ocimene and (Z)-β-ocimene were detected only in leaf tissues.

For the bark, the most abundant and highly significant compounds were α-pinene, camphene, limonene and neryl-acetate. It can be observed that α-pinene has an average relative amount higher in population A (53.4%) than B (15.5%). Conversely, in population B the relative amount of camphene (19.3%), limonene (48.8%) and neryl-acetate (17.2%) were in average higher than population A (3.4%, 22.1%, 2.4%, respectively). Finally, 2-bornene, α-phellandrene, (E)-β-ocimene, (Z)-β-ocimene, terpinolene, α-thujone, linalool, terpinel-4-ol and α-terpineol were absent in bark tissue. 

#### 2.3.2. Sesquiterpenes

In flowers, leaves and bark tissues 13 (93%), 12 (85%) and 7 (50%) sesquiterpenes out of a total of 14 were found (Figure 5).

In flower tissues, the most abundant and highly significant compounds were α-ylangene, β-caryophyllene, β-himachalene and δ-cadinene. The α-ylangene (9.7%), β-caryophyllene (18.2%) and δ-cadinene (13.1%) showed, on average, a higher relative amount in population A than in population B, where only a few traces of these compounds were present. The relative amount of β-himachalene was, on average, higher in population B (34.3%) than in population A (22.3%). Trans-bergamotene and β-cucumene were found only in the floral tissues.

In leaf tissues, α-copaene, β-caryophyllene, β-himachalene and α-curcumene were the most abundant and highly significant compounds. In population A, α-copaene (16.1%) and β-caryophyllene (13.5%) were higher than in population B, where these compounds were found in traces. In population B β-himachalene (29.2%) and α-curcumene (15.1%) were, on average, higher than in population A (16.3% and 5.4%, respectively). In addition, a sesquiterpene named Sesqui6_UNK was detected only in the flower tissues, and was more prevalent in population A (6.2%) than in population B, where it was found only in traces.

In bark tissues, the most abundant and highly significant compounds were α-ylangene, α-copaene, isoitalicene, β-caryophyllene and α-curcumene. The α-ylangene (4.65%), α-copaene (12.55%) and β-caryophyllene (8.13%) were higher in population A than in population B, where they were present just in traces. The isoitalicene and α-curcumene were higher in population B (7.65% and 23.24% respectively) than in population A (3.84% and 12.10%). Finally, cis-bergamotene, trans-bergamotene, Sesqui6_UNK, δ-cadinene, β-curcumene, guaiol and rosifoliol, were not found.

#### 2.3.3. Multivariate Analysis of Terpene Data

In flower tissue (Figure 6a, Appendix A), the two principal components explained 47.7% of the total variation among compounds. The most significant compounds were neryl-acetate, α-ylangene, δ-cadinene, β-caryophyllene and α-pinene (Dim1: 0.73, 0.67, 0.55, 0.53 and 0.48, respectively), explaining 32.5% of the total variation. The second component explained 14% of the total variation, with β-himachalene providing the major contribution (Dim2: 0.41).

In leaf tissue (Figure 6b, Appendix A), the two principal components explained about 46.60% of the total variation among compounds. The first principal component represented 34.6% of the total variation, which was explained by three monoterpenes (Dim1: neryl-acetate, 0.74; α-pinene, 0.68; β-myrcene, 0.46) and three sesquiterpenes (Dim1: Sesqui6_UNK, 0.63; β-caryophyllene, 0.52; α-curcumene, 0.46). For the second component, limonene was the main contributor (Dim2: 0.64), explaining 11.6% of the total variation.

In bark tissue (Figure 6c, Appendix A), the two principal components explained about 48.3% of the total variation among all compounds. The first principal component explained 34.7% of the total variation and the most significant compounds were α-pinene, isoitalicene, italicene, limonene, and α-curcumene (Dim1: 0.75, 0.65, 0.63, and 0.49, respectively). Moreover, the second component explained 13.6% of the total variation, with p-cymene observed to provide the higher contribution (Dim2: 0.68).

## 3. Discussion

### 3.1. Morphological Analysis

The morphological analysis suggested the possibility of distinguishing the two investigated populations, especially when cypselae and leaves were deeply assessed. In population A, a more pronounced leaf width was observed compared to population B, whilst in this latter, a more undulated leaf margin was observed. These differences in leaf characteristics suggest that population A was closer to *H. litoreum*, while population B was closer to *H. italicum*.

Other differences in morphological traits were found in the duplex-hairs density and the size of cypselae. Specifically, in this study, the cypselae in population A were found to be more elongated and with fewer duplex hairs than in population B. These observations were confirmed by the literature, where Pignatti [6] indicated that the cypselae of *H. litoreum* were bigger and with lower duplex hairs on the surface compared to those of *H. italicum*.

However, differences between populations were less clear when leaves and *capitula* were investigated. The literature suggested that the number of *capitula* per synflorescences was between 40 to 160 for *H. litoreum* [4], whilst in this study, the *capitula* per synflorescences collected in population A were between 10 and 24 (17.53 ± 6.55 mm). In addition, the length and width of leaves (21.53 ± 4.5 mm; 0.91 ± 0.2 mm) and *capitula* length (3.1 ± 0.24) were larger in population A compared to population B (16.82 ± 4.51 mm; 0.63 ± 0.15 mm; and 2.68 ± 0.37 mm, respectively), disagreeing with the literature [4].

The smaller size of the cypselae, the leaves and the *capitula* observed in population A than in population B may be due to a morphological variation caused by climatic conditions. This was suggested in the literature for *H. stoechas* [4] and other South African species [38]. Another hypothesis describes the presence of hybridized individuals, which may increase the complexity of the detection of specific individuals [1,4,39,40].

Another characteristic suitable for discriminating the two populations was the glands on the leaf’s abaxial surface [4,5]. In *H. litoreum,* the glands were expected to be sparser than in *H. italicum*, where they usually cover 70% of the *indumentum* [5]. This characteristic was found also in this study, where the glands on the leaf’s abaxial surface of the floral branch in population A were lower than those observed in population B.

### 3.2. Morphological Analysis

The samples of both populations were characterized by the external micromorphology of non-glandular and glandular trichomes, their histochemistry and distribution pattern on the investigated plant organs. The major differences referred to the density of the duplex hairs on the ovary/cypselae surfaces and the observation that the twin-celled structure was more evident in population B compared to population A. This observation was confirmed by the morphological analysis, as the density of the duplex hairs was a discriminating factor between the two populations. The duplex hair density was a discriminating factor for some species of *Helichrysum* [5,41] but it only partially described *H. italicum* and *H. litoreum* [4,6].

The long filamentous non-glandular trichomes were ubiquitous in all *Helichrysum* species examined, thus indicating morphological similarities within the genus. Based on external morphology and histochemistry, the glandular trichome morphotypes were similar to those previously described in other *Helichrysum* species such as *H. stoechas* [7], *H. splendidum* [30] and *H. italicum* [42]. Specifically, the club-shaped trichomes were found to be the main sites of production of terpene volatiles components, whereas the duplex trichomes on ovaries and cypselae were typically myxogenic structures secreting mucilaginous substances when moistened in water, favoring its absorption.

Moreover, in population A, cypselae were densely covered with epicuticular waxes in the form of small papillae, suggesting the findings of Salmeri et al. [35] for *H. litoreum*. By contrast, a substantial absence of small papillae on the surface of the cypselae was observed in population B. Again, Salmeri et al. [35] indicated that cypselae without epicuticular waxes can be found only for *H. italicum* subsp. *siculum*, while *H. italicum* subsp. *italicum* was not investigated. Therefore, since *H. italicum* subsp. *siculum* was found only in Sicily [5], the *Helichrysum* from population B can be assumed to belong to *H. italicum*. However, despite further literature reporting the presence of both *H. italicum* subsp. *italicum* and *H. italicum* subsp. *microphyllum* at Monte Capanne [33,43], the sole micromorphological analysis was not able to surely discriminate between these two species.

### 3.3. Terpene Analysis

In flowers, the α-pinene found in *H. litoreum* was higher than that observed for *H. italicum,* whilst few traces of neryl-acetate were observed. This pattern agreed with the findings of most current literature [29,31,44]. Furthermore, the values of α-pinene and neryl-acetate observed in flower tissues of *H. italicum* reflected those observed for the same population (B) around the Tuscan Archipelago [45,46,47], in some Italian internal areas [3,33], and Corsica and Sardinia [3,48,49,50].

Therefore, neryl acetate and a-pinene could be used as markers for *H. italicum* and *H. litoreum* in these areas. However, these compounds showed different dynamics in other areas. For instance, in Sicily [31,51], Crete [52], and Portugal [42], high concentrations of α-pinene were observed, and low or missing neryl acetate content, whilst in Croatia, their concentration was similar [53,54]. This behavior suggested that environmental conditions such as soil characteristics and climate may affect the concentration of these compounds, making challenging the discrimination between these two species on a large scale (i.e., across the Mediterranean basin).

In leaf tissues, the concentration of α-pinene and neryl acetate of both populations was similar to that observed in floral tissues. However, despite few studies analyzing leaf tissues, our results were found to be in agreement with the current literature, which indicated higher α-pinene concentrations in *H. litoreum* than in *H. italicum* [31,44]. By contrast, the concentrations of limonene and (E)-β-ocimene found in leaf tissues of population B and almost lacking in population A disagree with current literature, where limonene was found variable and (E)-β-ocimene was low or not present in the leaves of either *H. litoreum* and *H. italicum* [31,33,44,55]. Given this contrasting agreement with the literature, these compounds cannot be considered suitable for the chemotaxonomic identification of the two species.

In bark tissues, a high concentration of α-pinene was observed in population A, and very low neryl-acetate was found in population B. In addition, population B showed a higher relative limonene content than population A. To our knowledge, there is no literature reporting the pattern of monoterpenes in bark tissues of *Helichrysum;* however, the chemotaxonomic approach in bark tissue was performed for the genus *Picea* and *Abies* [22,56,57], suggesting that α-pinene, limonene and other monoterpenes in this plant compartment could be used as biochemical markers to identify different taxa. Overall, α-pinene, neryl acetate and limonene could be the most suitable markers in chemotaxonomic analysis to distinguish the two analyzed *Helichrysum* species.

Concerning the sesquiterpenes, in flower tissues of population A, the highest concentration was found for α-ylangene, δ-cadinene, and β-caryophyllene, while in population B these compounds were found only in traces and the highest concentration was found for β-himachalene. These results only partly agreed with the literature for *H. litoreum*, where a high content of β-caryophyllene was found, while α-ylangene, δ-cadinene and β-himachalene were almost lacking [31,44]. However, the low relative content of β-caryophyllene, δ-cadinene, and α-ylangene observed in *H. italicum* agreed with several scientific studies [6,33,47,53]. These dynamics suggested that the high presence of β-caryophyllene may be considered a possible marker for detecting *H. litoreum,* whilst the low presence of β-caryophyllene, δ-cadinene, and α-ylangene may detect the presence of *H. italicum.*

In leaf tissues, a very high concentration of β-caryophyllene was found in population A, while a high percentage of α-curcumene was observed in population B. For *H. litoreum*, studies confirmed the presence of a significant amount of β-caryophyllene and traces of α-curcumene in leaves [31,44]. For *H. italicum,* Bianchini et al. [58], and Giuliani et al. [33], reported low content of β-caryophyllene in Tuscany and Corsica, whilst Maggio et al. [31] found a higher concentration of β-caryophyllene in Sicily and Sardinia. The concentration of α-curcumene in *H. italicum* was observed to be more stable within different environments [59], resulting as a possible marker for detecting *H. italicum* rather than β-caryophyllene, which may be influenced by environmental conditions.

In bark tissues, β-caryophyllene was the main marker for population A, while α-curcumene was for population B. In population B, a higher percentage of the sesquiterpenes named italicene and isoitalicene compared to the other analyzed tissues was observed. As reported for monoterpenes, to our knowledge there are no studies analyzing sesquiterpenes in bark tissues of *Helichrysum,* and the analysis of sesquiterpenes could be considered a suitable approach to identifying specific populations [60,61]. For instance, Costa et al. [42] and Andreani et al. [50] reported the presence of italicene and isoitalicene in floral tissues of *H. italicum*, whilst several studies suggested the presence of these sesquiterpenes in other subspecies [31,48,62,63,64]. By contrast, these compounds were missing in *H. litoreum* [31,44]. Terpene analysis of bark tissue suggested that α-curcumene, isoitalicene, and italicene could be useful to identify *H. italicum*, while β-caryophyllene may be considered a possible marker of *H. litoreum*.

## 4. Materials and Methods

### 4.1. Study Area

Elba island (42°46′58′′ N; 10°17′11″ E) is the largest island of the Archipelago Toscano National Park (Livorno, Italy). The study area chosen was the only one where distinct populations of *H. italicum* subsp. *italicum* and *H. litoreum* were established [36,37,43].

The island landscape is heterogenous, with mountains in the western part, and plain areas in the central part of the island. The remaining areas are coastal and hilly, these latter characterized by large iron deposits. The climate is typically the Mediterranean, with warm and windy summers and mild winters. The coldest month is January, with a monthly mean temperature of 5.5 °C, while the warmest is August (27 °C). The long-term total annual rainfall average is less than 700 mm, with rainfall events mainly concentrated in autumn [65].

The two populations of *Helichrysum* spp. were collected in well-distinguished locations (Figure 7). *Helichrysum litoreum* was collected at Capo d’Enfola (site A), a coastal area (0–30 m a.s.l) in the north part of the island with south-west exposure. *Helichrysum italicum* subsp. *italicum* was collected in the eastern part of the island at Monte Capanne (site B), the highest peak of the island (1018 m a.s.l).

### 4.2. Plant Materials

A total of 60 plants of *Helichrysum* spp. were sampled on 18 June and 18 July 2020, at sites A and B, respectively. For each site, 30 plants were carefully selected randomly within the population to preserve the intra-site variability of the species. Then, for each plant, a portion of bark, vegetative shoot and full-blooming inflorescence was collected, following the protocol described by Squillace [27]. The collected samples were finally transported in refrigerated containers at 4 °C and then placed in the laboratory at −20 °C until terpene analysis. Moreover, for each site, 15 plants from each population were sampled to obtain *exsiccata* for morphological analysis.

### 4.3. Morphological Analysis

In this study, three variables were further analyzed compared to the 24 variables currently applied for the morphological description of the *Helichrysum* (Table 2). Specifically, the glandular *indumentum* of the leaf abaxial side (ID05); the axillary fascicles of the floral stem (ID21); and the ratio between synflorescence length and the number of *capitula* per synflorescence (ID27) were evaluated. In addition, for ID05, ID14, ID17, and ID20, a quantitative approach rather than only qualitative information was reported. For ID05 and ID20, the glandular *indumentum* of the proximal part of the leaf’s abaxial side was also investigated.

Morphological measurements were performed using digital images on *exsiccata* at different resolution (i.e., 15×; 50×; 240×). All images were obtained using a DinoLite digital microscope (mod. AM4113ZT) and processed with the DinoCapture software (ver. 2.0) (Table 2). For all characteristics, where possible, the mean of three to six measurements per specimen was used.

### 4.4. Micromorphological Analysis

Phytochemical analysis was carried out on cortex, vegetative shoot and full-blooming inflorescence by extraction through sonication and maceration in heptane. The extract was injected into GC-MS (‘Agilent model 7820A’) equipped with a single quadrupole mass detector (model 5977E). The adapted analytical conditions were injector temperature 260 °C in spitless mode; 50 m HP-Innowax column, having 0.2 mm i.d. and 0.4 μm absorbent film thickness; He carrier gas at a flow rate of 1.2 mL/minute in constant flow mode. The mass detector was set in scan mode in the m/z range 29–350, at a rate of 3 scans per second. GC-MS chromatograms were processed with “Agilent Mass Hunter Quantitative Analysis” software through deconvolution, dynamic background subtraction and comparison of the spectra of the separated peaks with those reported in the 2011 NIST library.

### 4.5. Terpene Analysis

Terpene analyses were performed three weeks from the sample collection of bark, vegetative shoot, and full-blooming inflorescence. The analysis was carried out by sonicating and macerating the samples in heptane. The samples were weighed in 1.5 mL vials (i.e., 0.01 g of bark, 0.02 g of the central part of the vegetative shoot and 0.02 g of full-blooming *capitula*) and extracted with 1 mL of heptane. Then, samples were vortex mixed and incubated in an ultrasonic bath (BandelinSonorex Super RK 102H) for 15 min three times and successively incubated at 30 °C with a rotary shaker at 10 revolutions per minute for 24 h. The extracted samples were centrifuged at 3000 rpm for 15 min. Finally, 1 μL of the supernatant was injected into GC-MS (Agilent model 7820A) equipped with a single quadrupole mass detector (model 5977E). The adapted analytical conditions were injector temperature 260 °C in spitless mode; 50 m HP-Innowax column, having 0.2 mm i.d. and 0.4 μm absorbent film thickness; helium carrier gas at a flow rate of 1.2 mL/minute in constant flow mode. The mass detector was set in scan mode in the m/z range 29–350, at a rate of 3 scans per second. GC-MS chromatograms were processed with “Agilent Mass Hunter Quantitative Analysis” software through deconvolution and dynamic background subtraction. Identification of compounds was performed by comparing their mass spectra with those reported in the National Institute of Standards and Technology (NIST 11, Gaithersburg, MD, USA) mass spectra library. “Tentative” identification was based on Kovats retention indices (KI) and identification was confirmed by comparison of KI and mass spectra with those of available standards. The relative amount (percentages) of each monoterpene was expressed as a percentage of total monoterpenes (monoterpene profiles), while the relative amount (percentages) of each sesquiterpene was expressed as a percentage of total monoterpenes (terpene profiles).

### 4.6. Statistical Analysis

Analysis of variance (ANOVA) was carried out both on morphometric and terpene data. The analysis was performed using R Studio v. 1.4 [67] and SYSTAT v 13.4 (Systat Software, Inc., San Jose, CA, USA, 2021). The distribution of qualitative morphological characters was analyzed by the chi-square test, whilst normality and homoscedasticity of quantitative variables were analyzed with Shapiro–Wilk and Levene test. Variables that did not fulfil ANOVA assumptions, even after transformation, were analyzed by Kruskal–Wallis’s test or Welch’s test.

Principal component analysis (PCA) was also carried out using R Studio with FactorMineR and Factorextra packages [68,69] for both morphological and terpene data. The correlation between significant ANOVA morphological traits and terpene compounds was provided by coefficients of Spearman and Pearson, respectively. The highly correlated variables (r > |0.95|) were excluded from analyses.

## 5. Conclusions

The primary objective of this work was to discriminate between the two wild populations of *Helichrysum* spp. utilizing morphological and phytochemical approaches. Morphological analysis revealed that leaf width and margin, the density of the glandular structure in the abaxial leaves *indumentum* and the size of cypselae were the main traits discriminating between the two species. Micromorphological analysis indicated the waxy microstructures and the density of the duplex-hair on the cypsela as the main characteristics discriminating between *H. litoreum* to *H. italicum*.

Terpene analysis revealed that, among all monoterpenes, the high content of neryl-acetate could make it the most suitable compound to identify *H. italicum*, while high contents of α-pinene characterize *H. litoreum*. Among sesquiterpenes, the presence of a high relative percentage of α-curcumene, isoitalicene and italicene could be the most stable and discriminating compounds for *H. italicum*, while the high relative content of β-caryophyllene could be a possible marker for *H. litoreum.* Finally, bark tissue revealed its suitability in chemotaxonomic analysis, since the terpene profile of bark tissue provided more stable compounds that could be used as biochemical markers.

However, although our analysis reported numerous pieces of evidence, future studies are needed. First, an in-depth genetic analysis of *Helichrysum* populations would help to define the species distribution in the Tuscan Archipelago territory. In addition, to further refine the chemotaxonomic analysis, our future studies will analyze terpene profiles of seedlings of vegetatively propagated clones from different Italian populations of *H. litoreum* and *H. italicum* grown in an experimental plantation under the same environmental conditions.

## Figures and Tables

**Figure 1 plants-11-01719-f001:**
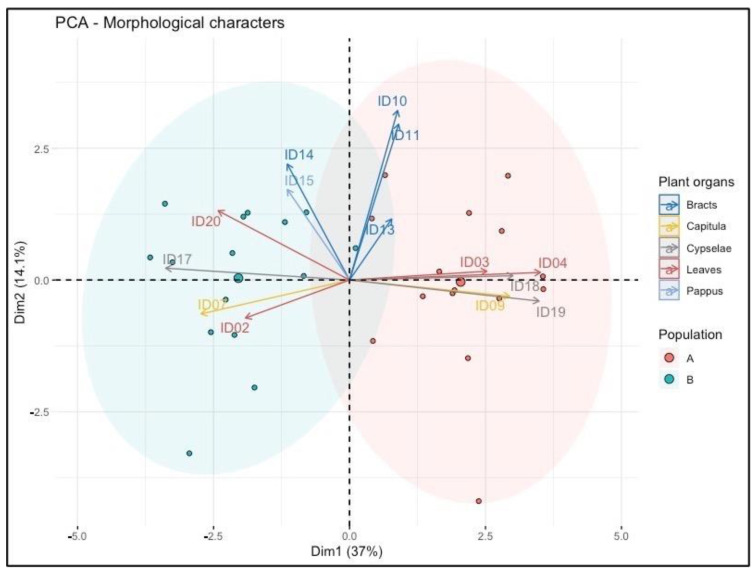
Principal component graph of morphological characters based on the first two components. *H. litoreum* is reported in population A, and *H. italicum* subsp. *italicum* in population B.

**Figure 2 plants-11-01719-f002:**
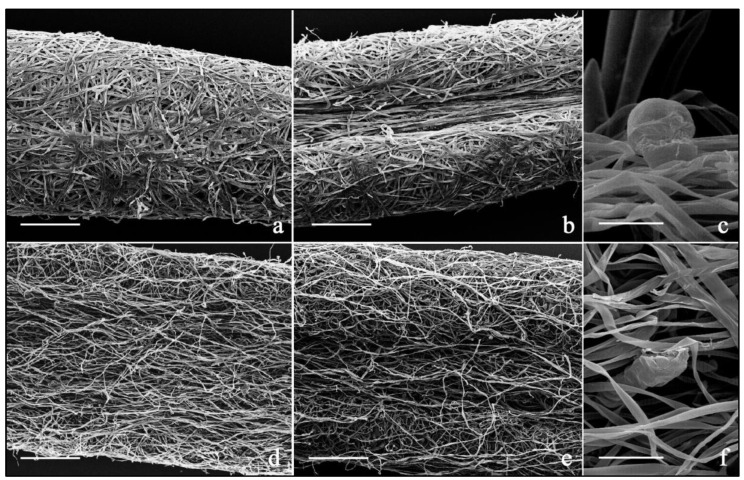
(**a**–**f**). SEM micrographs showing the distribution pattern of trichome morphotypes in the investigated *Helichrysum* populations. (**a**–**c**). Population A samples: leaf adaxial (**a**) and abaxial surfaces (**b**) covered with long filamentous non-glandular trichomes; particular of the apical cells of a club-shaped glandular trichome (**c**). (**d**–**f**). Population B samples: leaf adaxial (**d**) and abaxial surfaces (**e**) covered with long filamentous non-glandular trichomes; particular of the apical cells of a club-shaped glandular trichome (**f**). Scale bars = 200 μm (**a**,**b**,**d**,**e**); 25 μm (**c**,**f**).

**Figure 3 plants-11-01719-f003:**
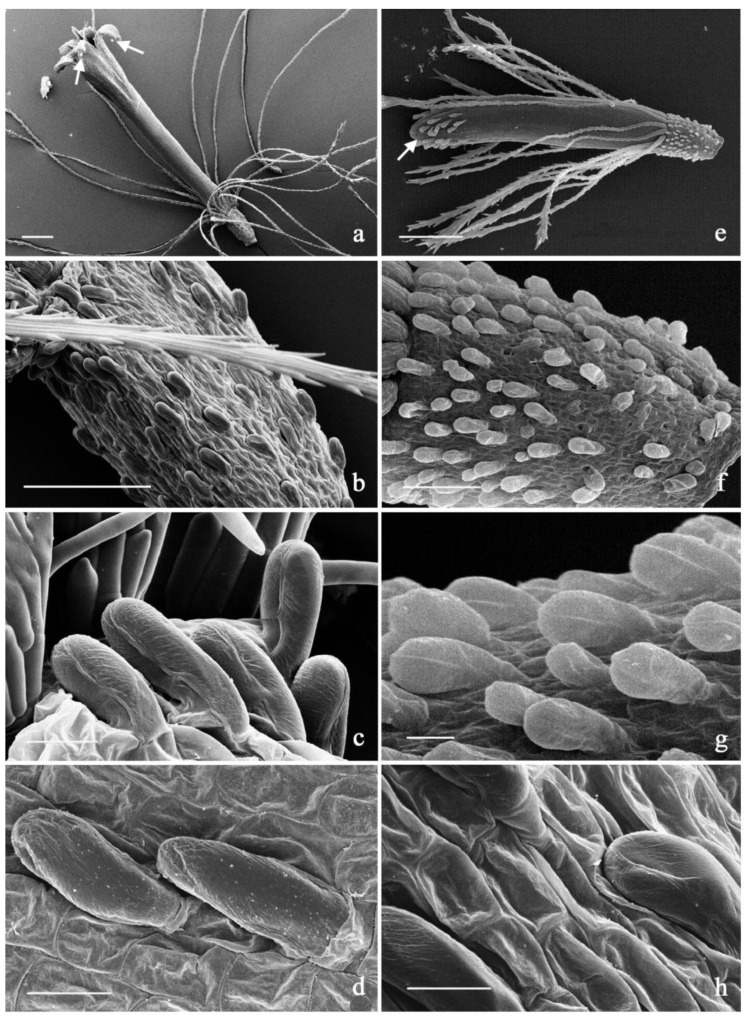
(**a**–**d**). SEM micrographs showing the distribution pattern of trichome morphotypes in the investigated *Helichrysum* A population (Capo d’Enfola). (**a**). General view of a mature floret. (**b**). Particular the ovary with duplex glandular trichomes. (**c**,**d**). Details of the duplex glandular trichome morphotype. (**e**–**h**). SEM micrographs showing the distribution pattern of trichome morphotypes in the investigated Helichrysum B population (Monte Capanne) (**e**). General view of a mature floret. (**f**). Particular the ovary with duplex glandular trichomes. (**g**,**h**). Details of the duplex glandular trichome morphotype. Scale bars = 200 μm (**a**,**e**); 80 μm (**f**); 25 μm (**c**,**d**,**g**); 10 μm (**b**,**h**).

**Figure 4 plants-11-01719-f004:**
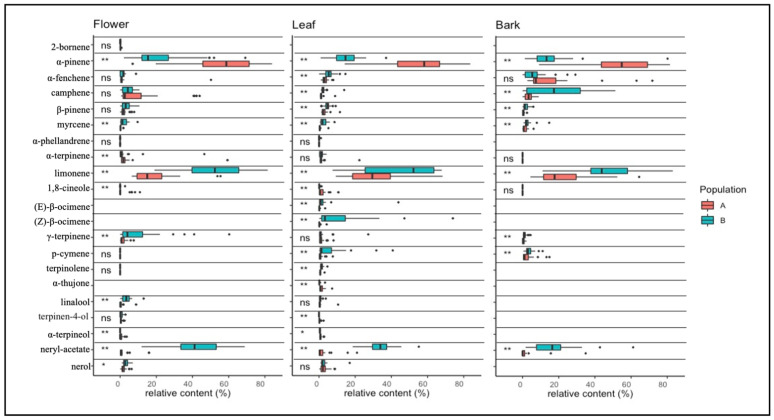
Boxplot and ANOVA of monoterpenes in flower, bark, and leaf tissues. Significance ANOVA code are: (ns) for *p* > 0.05; (*) for 0.05 < *p* >0.01; and (**) for *p* < 0.01. *H. litoreum* is reported in population A, and *H. italicum* subsp. *italicum* in population B.

**Figure 5 plants-11-01719-f005:**
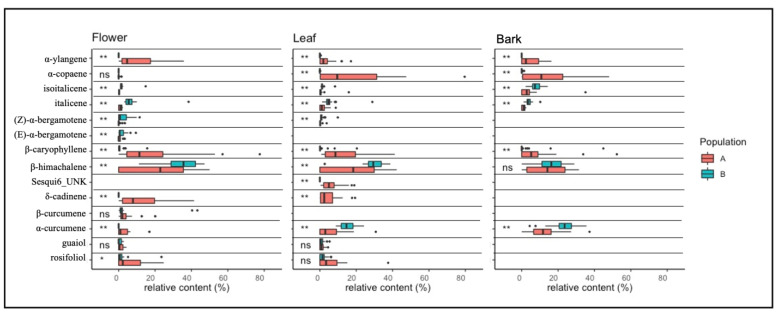
Boxplot and ANOVA of sesquiterpenes in flower, bark, and leaf tissues. Significance ANOVA codes are: (ns) for *p* > 0.05; (*) for 0.05 < *p* > 0.01; and (**) for *p* < 0.01. *H. litoreum* is reported in population A, and *H. italicum* subsp. *italicum* in population B.

**Figure 6 plants-11-01719-f006:**
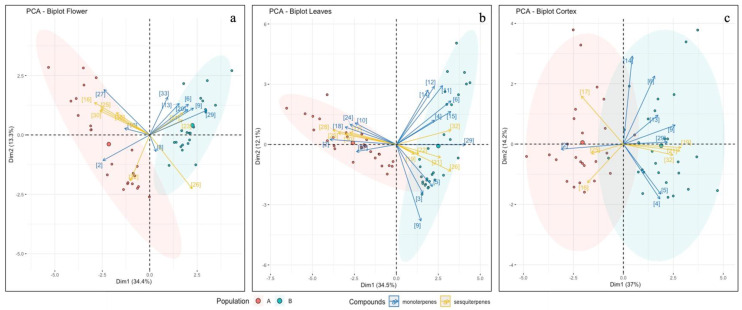
Principal component graph of monoterpenes and sesquiterpenes in flowers (**a**), leaves (**b**) and bark tissues (**c**) based on the first two principal components. Number on arrows represents compounds: (2) α-pinene; (3) α-fenchene; (4) camphene; (5) β-pinene; (6) myrcene; (8) α-terpinene; (9) limonene; (10) 1,8-cineole; (11) (E)-β-ocimene; (12) (Z)-β-ocimene; (13) γ-terpinene; (14) p-cymene; (15) terpinolene; (16) α-ylangene; (17) α-copaene; (18) α-thujone; (19) isoitalicene; (20) linalool; (21) italicene; (22) (Z)-α-bergamotene; (23) (E)-α-bergamotene; (24) terpinel-4-ol; (25) β-cariophillene; (26) β-himachalene; (27) α-terpineol; (28) Sesqui6_UNK; (29) neryl-acetate; (30) δ-cadinene; (32) α-curcumene; (33) nerol; (35) rosifoliol. *H. litoreum* is reported in population A, and *H. italicum* subsp. *italicum* in population B.

**Figure 7 plants-11-01719-f007:**
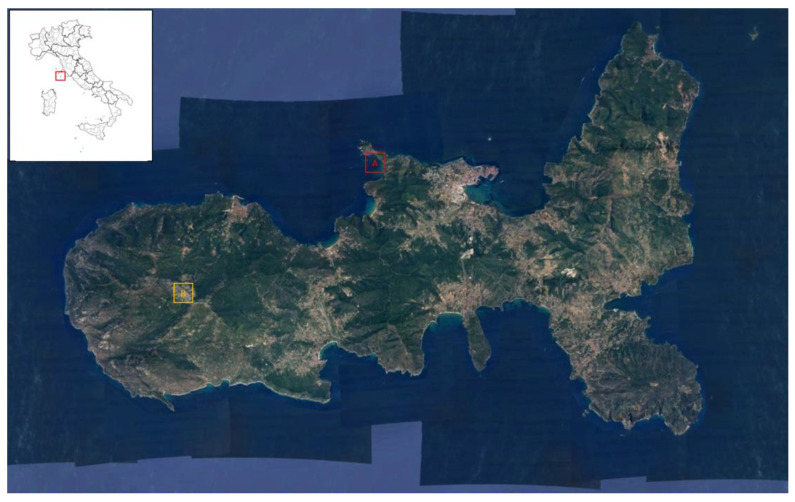
Location of the two study sites on Elba Island: the coastal (Capo d’Enfola A) and the mountain (Monte Capanne, B) sites.

**Table 1 plants-11-01719-t001:** Morphological characters of two wild populations of *Helichrysum* spp. (*H. litoreum* and *H. italicum* subsp. *italicum*) from the Tuscan Archipelago (Central Italy). ID is the Identification number of the single morphological variables; A and B represent the population of the two species, respectively.

ID.	Morphological Variables	A	B	*p*-Value	Sig.
01	Axillary leaf fascicles (vegetative stem) ^a^	1.82 ± 0.39	1.94 ± 0.23	0.85	ns
02	Leaf margin undulation ^a^	1.08 ± 0.27	1.63 ± 0.19	0.00	**
03	Caulinar leaf length ^b^	21.53 ± 4.50	16.82 ± 4.51	0.00	**
04	Caulinar leaf width ^b^	0.91 ± 0.20	0.63 ± 0.15	0.00	**
05	Glandular *indumentum* of leaf abaxial side (vegetative stem) ^c^	12.24 ± 4.84	12.31 ± 7.44	0.77	ns
06	Synflorescence length ^c^	23.73 ± 5.27	26.69 ± 7.40	0.99	ns
07	Number of *capitula* per synflorescence ^c^	17.53 ± 6.55	33.04 ± 13.43	0.02	*
08	*Capitulum* length ^c^	4.26 ± 0.51	4.16 ± 0.34	0.86	ns
09	*Capitulum* width	3.09 ± 0.24	2.68 ± 0.37	0.00	**
10	Outermost involucral bract lenght	1.16 ± 0.07	0.97 ± 0.06	0.00	**
11	Outermost involucral bract width ^b^	0.57 ± 0.12	0.45 ± 0.16	0.03	*
12	Innermost involucral bract length ^b^	3.66 ± 0.34	3.59 ± 0.71	0.52	ns
13	Innermost involucral bract width	0.64 ± 0.09	0.59 ± 0.09	0.01	*
14	Glandular *indumendum* of Innermost involucral bract	44.44 ± 9.11	27.43 ± 7.76	0.00	**
15	Pappus setae length ^b^	2.89 ± 0.30	3.07 ± 0.24	0.00	**
16	Shape of pappus apical cells ^a^	1.03 ± 0.16	1.16 ± 0.37	0.46	ns
17	Cypsela duplex hair density ^c^	1.53 ± 0.56	4.42 ± 1.72	0.02	*
18	Cypsela length	0.73 ± 0.10	0.58 ± 0.09	0.00	**
19	Cypsela width ^b^	0.37 ± 0.07	0.23 ± 0.07	0.00	**
20	Glandular *indumendum* of leaf abaxial side (floral stem) ^c^	15.10 ± 5.81	26.81 ± 13.13	0.02	*
21	Axillary leaf fascicles (floral stem) ^a^	1.35 ± 0.48	1.25 ± 0.44	0.40	ns
22	Caulinar leaf length/caulinar leaf width ^c^	24.20 ± 5.25	27.44 ± 7.10	0.01	*
23	*Capitulum* length/*capitulum* width ^b^	1.41 ± 0.22	1.57 ± 0.20	0.29	ns
24	Outermost involucral bract length/ Outermost involucral bract width	2.05 ± 0.49	2.51 ± 1.71	0.57	ns
25	Innermost involucral bract length/ Innermost involucral bract width ^b^	5.80 ± 0.90	6.22 ± 1.28	0.07	ns
26	Cypsela length/Cyspela width ^c^	2.00 ± 0.34	2.75 ± 0.38	0.02	*
27	Synflorence length/ Number of *capitula* per synflorescence ^b^	1.47 ± 0.37	0.92 ± 0.16	0.00	**

Values are means with standard deviation (*n* = 30). Significance ANOVA code are: (ns) for *p* > 0.05; (*) for 0.05 < *p* > 0.01; and (**) for *p* < 0.01. ^a^ Chi-squared test applied, ^b^ Welch’s test applied, ^c^ Kruskal–Wallis’s test applied.

**Table 2 plants-11-01719-t002:** List of morphological characters used in the morphometric analysis, with the indication of the unit of measure and the references.

ID.	Morphological Characters	Unit of Measure	References
01	Axillary leaf fascicles (vegetative stem)	1 absence; 2 presence	[5,41]
02	Leaf margin undulation	1 most without; 2 some leaves, 3 most leaves	[5]
03	Caulinar leaf length	mm	[4,5]
04	Caulinar leaf width	mm	[4,66]
05	Glandular *indumentum* of leaf abaxial side (floral stem)	n/0.64 mm^2^	[4,5]
06	Synflorence length	mm	[5]
07	Number of *capitula* per synflorescence	mm	[4,5]
08	*Capitulum* length	mm	[4,5]
09	*Capitulum* width	mm	[4,5]
10	Outermost involucral bract lenght	mm	[4,66]
11	Outermost involucral bract width	mm	[4,66]
12	Innermost involucral bract length	mm	[40,66]
13	Innermost involucral bract width	mm	[40,66]
14	Glandular *indumendum* of Innermost involucral bract	n/0.8 mm^2^	[40,66]
15	Pappus setae length	mm	[5]
16	Shape of pappus apical cells	1 acute; 2 obtuse	[4]
17	Cypsela duplex hair density	n/0.04 mm^2^	[4]
18	Cypsela length	mm	[5]
19	Cypsela width	mm	[5]
20	Glandular *indumendum* of leaf abaxial side (floral stem)	n/0.64 mm^2^	This work
21	Axillary leaf fascicles (floral stem)	1 absence; 2 presence	This work
22	Caulinar leaf length/caulinar leaf width	Ratio: 1/2	[4,5]
23	*Capitulum* length/*capitulum* width	Ratio: 6/7	[4,5]
24	Outermost involucral bract length/ Outermost involucral bract width	Ratio: 8/9	[5]
25	Innermost involucral bract length/ Innermost involucral bract width	Ratio: 10/11	[5]
26	Cypsela length/Cyspela width	Ratio: 15/17	[5]
27	Synflorence length/ Number of *capitula* per synflorescence	Ratio: 4/5	This work

## Data Availability

Not applicable.

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
