# Peer review of "Terpene Profiles Composition and Micromorphological Analysis on Two Wild Populations of Helichrysum spp. from the Tuscan Archipelago (Central Italy)"

_plants, 2022, doi:10.3390/plants11131719_

Round 1
Reviewer 1 Report
The results of mathematical data processing must be presented strictly according to the rules. Including numerical values in the tables should be rounded to tenths, maximum - to hundredths.
If possible, these species should be compared in another territory of Italy, and not only.
How successful are scanning microscope methods in determining or refining the species of each species? It is necessary to indicate clear morphological differences between species.
Author Response
Please see the file attached.

Reviewer 2 Report
In general, the article as it stands is suitable for publication in Plants. However, two points need clarification.
Was terpinel-4-ol or maybe terpinen-4-ol among the identified compounds shown in Figure 4?
In the "Reference" section, Latin plant names should be written in italics. Please check.
Reviewer 3 Report
Dear Colleagues, this paper adds value to the field of research on the morphological, micromorphological and chemical characterization of two wild populations of Helichrysum spp. One of the species is endemic, so the results obtained could be used for better conservation of the species but also for sustainable exploitation programs. I believe that the presentation of the results needs to be improved so that the text to be clearer and easier to understand given that this chapter appears before the chapter on materials and methods.
I have some specific comments:
L72 '' for the treatment of Helichrysum spp. '' I did not understand what it refers to, maybe it should be reformulated.
L88 '' revealed that the most significant traits '' please specify if it refers to the characters that differentiate the two species / populations.
L-89-90 please specify what is ID.
In table 1 please specify what ID, A, B represent. Also in the table please be careful that the lower case letters (a, b, c) corresponding to the statistical tests are not included, please include them.
I think the title of the Table 1 should be changed as follows: Morphological characters of two wild populations of Helichrysum spp. (H. italicum subsp. Italicum and H. litoreum) from the Tuscan Archipelago (Central Italy).
Below the table should be included: Values ​​are means with standard deviation (n = 30). Significance ANOVA code are: (ns) for p> 0.05; (*) for 0.05  0.01; and (**) for p <0.01.
I also think that '' Sign '' could be replaced by '' Sig. ''
I think that in the description of figures 1, 4, 5, 6 the studied species should be specified.
In the supplementary material the title of the tables must be entered above. For Table 5 you should specify what A and B represent.
Reviewer 4 Report
See attached document.
